# Associated factors with Premenstrual syndrome and Premenstrual dysphoric disorder among female medical students: A cross-sectional study

Vy Dinh Trieu Ngo[1‡]*, Linh Phuong Bui[2,3‡], Long Bao Hoang[3,4], My Thi Tra Tran[5], Huy Vu Quoc Nguyen[6], Linh Manh Tran[6‡], Tung Thanh Pham[3,7,8‡]

1 Tam Anh TP. Ho Chi Minh General Hospital, Ho Chi Minh, Vietnam, 2 Department of Nutrition, Harvard T. H. Chan School of Public Health, Boston, Massachusetts, United States of America, 3 Research Advancement Consortium in Health, Hanoi, Vietnam, 4 Institute of Gastroenterology and Hepatology, Hanoi, Vietnam, 5 Department of Psychiatry, Hue University of Medicine and Pharmacy, Hue University, Hue, Vietnam, 6 Department of Obstetrics and Gynaecology, Hue University of Medicine and Pharmacy, Hue University, Hue, Vietnam, 7 Department of Epidemiology, Harvard T.H. Chan School of Public Health, Boston, Massachusetts, United States of America, 8 Department of Physiology, Hanoi Medical University, Hanoi, Vietnam

‡ VDTN and LPB contributed equally to this work as co-first authors. LMT and TTP contributed equally to this work as co-last authors.
* ngodinhtrieuvygl@gmail.com, vyndt@hcm.tahospital.vn

**Data Availability Statement:** All relevant data are within the paper and its Supporting information files.

## Abstract

### Aim

The study aimed to determine potential risk factors associated with Premenstrual Syndrome and Premenstrual Dysphoric Disorder.

### Methods

Three hundred two female student participants who were 18–45 years old completed a questionnaire including demographic characteristics, lifestyle factors, and a Vietnamese Premenstrual Syndrome Screening Tool. We then followed up participants during at least two menstrual cycles using the Daily Record of Severity of Problems. The Premenstrual Syndrome and Premenstrual Dysphoric Disorder diagnosis was established using The Carolina Premenstrual Assessment Scoring System, based on the American College of Obstetrics and Gynecology and Diagnostic and Statistical Manual of Mental Disorders.

### Results

According to the Carolina Premenstrual Assessment Scoring System, 35 out of 302 students (11.6%; 95%CI: 8.2–15.7%) met the diagnosis of PMS (31 students) or PMDD (4 students). We found that age at menarche (PR = 0.77, 95%CI: 0.63–0.96), having negative Rh blood type (PR = 4.43, 95%CI: 1.95 to 10.08), being moderately depressed or higher (PR = 2.81, 95%CI: 1.24 to 6.36), and consuming caffeine more than three times per week

**Funding:** The study had been supported by the Research Advancement Consortium in Health and Hue University of Medicine and Pharmacy, Vietnam (https://www.reach.edu.vn/pmspmdd.html). The funders had no role in study design, data collection and analysis, decision to publish, or preparation of the manuscript.

**Competing interests:** Vy. D. T. Ngo has received a research grant from Research Advancement Consortium in Health (REACH) which Linh P. Bui, Long B. Hoang and Tung T. Pham are founders and members of REACH (a non-profit entity). Linh P. Bui, Long B. Hoang and Tung T. Pham did not receive any payment or compensation from this position at REACH. They provided consultancy on study design, data collection and analysis, and preparation of the manuscript. However, REACH had no role in the decision to publish this study, and this final decision belongs to the funded research team. This does not alter our adherence to PLOS ONE policies on sharing data and materials.

were statistically associated with having Premenstrual Syndrome or Premenstrual Dysphoric Disorder after adjusting for other variables.

## Conclusion

The prominent risk factors for Premenstrual Syndrome and Premenstrual Dysphoric Disorder were negative Rhesus blood type, menarche age, caffeine consumption, and self-reported depression.

## Introduction

Premenstrual syndrome (PMS) and Premenstrual dysphoric disorder (PMDD) are two premenstrual disorders that have been reported in many countries and have gradually become predominant concerns [1–3]. In a meta-analysis including 17 studies, prevalence of PMS ranged from 12% (in France) to 98% (in Iran), with a pooled estimate of 47.8% (95% CI: 32.6–62.9) [4–6]. PMDD is a severe disorder of PMS affected 3–8% of reproductive age women verified by daily record of severity problems (DRSP) [7].

In 2003, Steiner developed a Premenstrual syndrome screening tool (PSST) for rapid screening of PMS and PMDD that only requires subjects to answer questions once instead of monitoring menstrual cycles [8]. Because of a relatively high validity [8–10], PSST has been recommended as PMS / PMDD screening tool by International Society for Premenstrual Disorders (ISPMD) [10–13]. However, a PMS/PMDD definitive diagnosis requires the confirmation of fluctuating symptoms during the pre- and post-menstrual phases for at least two positively symptomatic menstrual cycles according to 5th Diagnosis and Statistics Manual of Mental Disorders (DSM-V) [10, 14, 15]. Among many other validated techniques, the most commonly used and accepted tool for diagnosing PMS/PMDD is DRSP. In order to efficiently summarize results from DRSP, Eisenlohr-Moul et al. developed an algorithm called the Carolina Premenstrual Assessment Scoring System (C-PASS) [12, 16] that analyzes DRSP in standardized manner.

The influence of PMS and PMDD on women's quality of life is well documented in the literature. The etiology of these disorders involve many factors such as genetics, genomics, developmental exposures, or comorbidities; however, the exact mechanisms of action are poorly understood [17–19]. In particular, the menstrual variation of the reproductive hormones (estrogen and progesterone), neurotransmitters (serotonin, noradrenaline, gamma-Aminobutyric acid), and inflammatory factors (prostaglandins) related to PMS/PMDD have been reported [20–23]. Moreover, many authors also suggest that interpersonal relationships and cooperation, stress, biological factors (genetics, length of menses, and pregnancies), and lifestyle exposures (dietary habits, physical exercise, or stimulants) could be potential risk factors for PMS/PMDD.

As pathogenesis is still poorly understood, treatment focuses mainly on mitigating symptoms via using medication. The first-line treatment is to use serotonergic antidepressants (Selective serotonin reuptake inhibitors (SSRIs)) to modulate serotonin level. Other drug options are GnRH agonists or estrogens, which are considered endocrine therapies to suppress ovulation. However, these drug treatment carry worrisome side effects; the patient needs to consult with physicians before initiating treatment [12, 24]. Besides, American Association Family Physician (AAFP) emphasized that eliminating modifiable risk factors can improve the

severity of PMS and PMDD [10, 25]. Moreover, the investigation of modifiable and non-modifiable risk factors plays a vital role in understanding the underlying mechanisms.

In Vietnam, PMS and PMDD prevalence as well as its risk factors have not been well documented. We initially reported the PMS and PMDD proportions according to DSM-V criteria in the Vietnamese female students as 9.9% and 1.0%, respectively [26]. This study aimed to determine potential risk factors associated with PMS/PMDD among the Vietnamese female student aged 18–45 via C-PASS or PSST.

## Methods

### Ethical approval

This study was approved by the Institutional Review Board of Hue University of Medicine and Pharmacy with the registration number: *H2019/003*. We obtained verbal and written consent from the participants after explaining the relevant information such as study context, objectives, data collection procedure.

### Study design and setting

A cross-sectional study among female students at Hue University of Medicine and Pharmacy (HueUMP), Hue, Vietnam was conducted from December 2018 to October 2019. In this study, we recruited a convenience sample of Vietnamese female students between the ages of 18 to 45 years who self-reported having regular menstrual cycles ranging from 24 to 35 days. We excluded participants who were taking hormonal therapy that could affect menstrual status, suffering from psychological conditions, endocrine diseases, endometriosis, or had severe/chronic medical diseases that required critical care.

### Sample size and data collection

The sample size was estimated based on the projected PSST sensitivity of 80%, and the estimated PMS/PMDD prevalence of 30% [27], with a 10% margin of error and a 95% confidence interval [14, 28, 29]. Our study calculated the required minimal sample size of 207. Predicted about 40% of participants would refuse to enter the follow-up and 10% would be lost-to-follow-up during the two menstrual cycles, the targeted total sample was at least 310 participants. We used convenience sampling to recruit the participants and to gain quality responses while minimizing the loss to follow up rate.

There were two major phases including a *Baseline assessment* and *a Follow-up phase* (Fig 1). At first, all interested students filled in a self-reported screening questionnaire to determine their eligibility. Research staff then invited eligible students to fill out the baseline questionnaires (including PSST and demographic information), and took anthropometric measurements. We measured weight by using *OMRON* weight—model HBF-214 [30] and height according to WHO guideline [31].

All eligible participants were then invited to enter the follow-up phase, in which at least two menstrual cycles would be monitored to detect PMS/PMDD symptoms. The participants reported daily levels of 25 symptoms via *Daily record of severity of problems (DRSP)* distributed through an online Google form. The research team built a community on the social network to remind and address common questions of the participants. Participants also received reminders directly via their preferred communication channels such as messenger, email, or a phone call. At the end of the follow-up phase, the participants were also asked to complete a second (re-test) PSST.

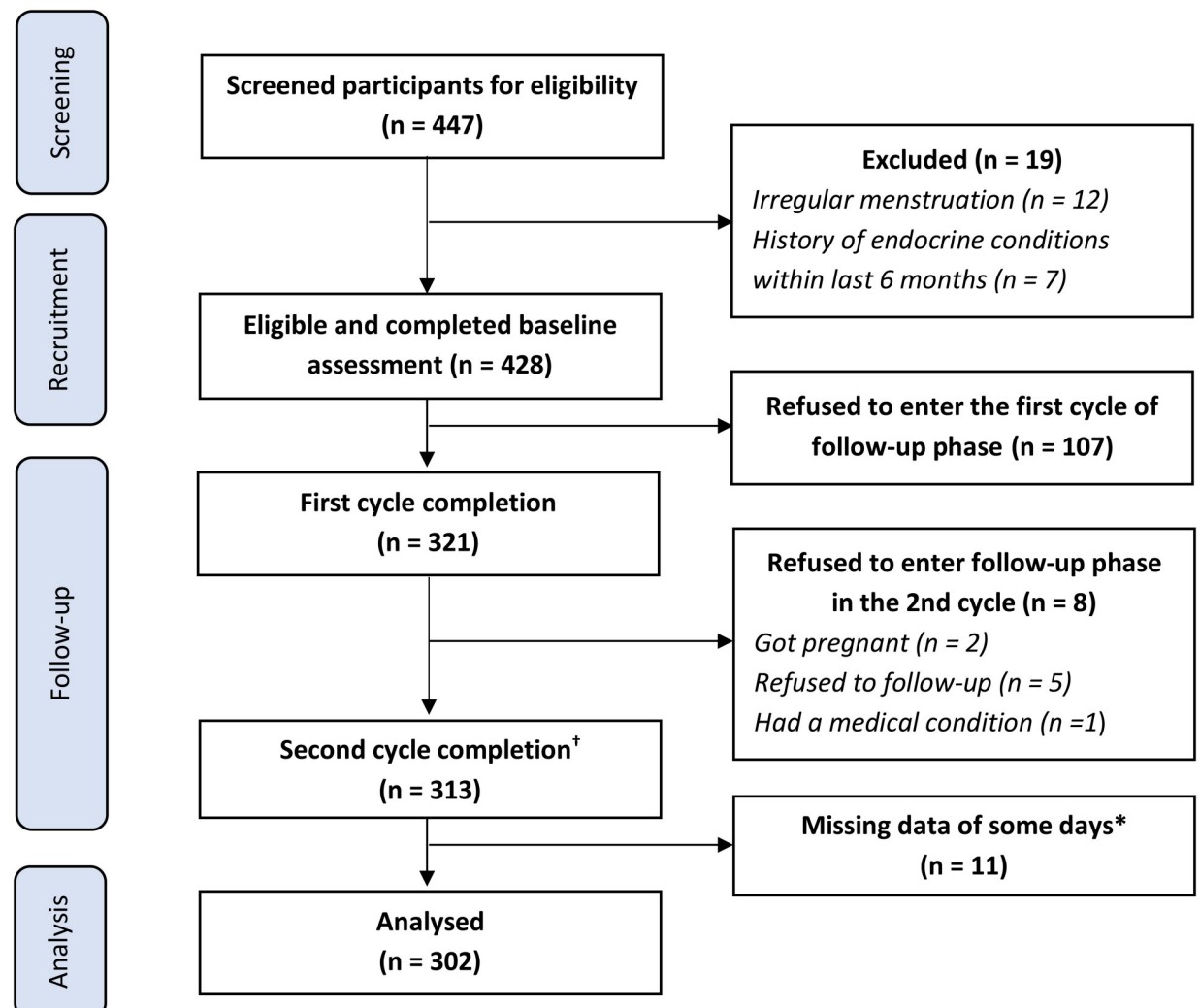

**Fig 1. Flow chart of participant recruitment and retention.** *The participants could report the score of symptoms of each day in DRSR (Daily Record of Severity Problems) within three days from the submission day. We excluded participants if they missed more than 3 days in a row or repeatedly missed the report of any day up to 5 times. † There were 11 participants would like to be followed and completed the third cycle.

### Research instruments

The questionnaire at baseline assessment consisted of items that could be related to PMS/PMDD including menstrual status (age of menarche, number of menstrual days and cycle days); socio-demographic factors; psychological history of first degree relatives; *PSST* to screening PMS/PMDD [8]; the *Patient Health Questionnaire (PHQ-9)* [32] to screen for depressive symptoms, lifestyle factors such as alcohol consumption, caffeine intake, smoking status. Physical activity was assessed based on the *International Physical Activity Questionnaire Short Form (IPAQ-SF))* [33]. In the follow-up phase, the symptoms of PMS/PMDD were reported according to the *DRSP*.

In 2003, Steiner et al. developed PSST, which consists of 19 items subdivided into two domains (manifestations and functional impact of PMS), as a screening tool for PMS [8]. Each item of PSST was scored using a 4-point Likert scale (0 = absent; 1 = mild; 2 = moderate; 3 = severe) [8]. The first domain had 14 symptoms including four core symptoms and ten

other symptoms regarding decreased interest in daily activity, behavioral signs and physical symptoms. The five variables of second domain consists of the interference with daily activity [8]. A positive participant for PMDD would need to show (i) at least five symptoms of the first domain with a score $\geq 2$; and (ii) at least one of the first four core symptoms must be rated as severe (score = 3); and (iii) at least one of five variables was rated as severe (score = 3) in the second domain. Positive PMS screening would require the same (i), (ii), and (iii) criteria as PMDD; however, the level of the four core symptoms and functional impacts (second domain) ranged from moderate to severe [8]. When compared with the diagnostic criteria of the DSM-V, PSST showed high sensitivity (66.3% to 79.0%) and varied specificity (33.3% to 85.6%) [9, 34].

Regarding DRSP, DSM-V states that daily ratings are essential for a diagnostic confirmation of PMDD [35, 36]. DRSP was designed with 21 separate psychological and physical symptoms as well as an additional three items to describe specific types of impairment in functioning caused by the symptoms [36]. The ratings on the DRSP are to be made daily by the subject throughout her menstrual cycle, on items with a 6-point severity scales. The levels of severity on the DRSP are 1—Not at all, 2—Minimal, 3—Mild, 4—Moderate, 5—Severe, 6—Extreme [10]. Using DRSP data of at least two consecutive menstrual cycles, we applied CPASS to confirm positive PMS/PMDD cycle [16]. The agreement of C-PASS diagnosis with the expert clinical diagnosis was reported at 98% [16]. A detailed algorithm of CPASS is available elsewhere [37]. In brief, a symptom was considered positive if the mean score of 10 days after menstruation (postmenstrual phase) is lower than the one of 7 days before menstruation (premenstrual phase) and the difference must be more than 30% of the woman's "range of scale used" (calculated by subtracting 1 from the woman's maximum rating across all DRSP responses in all cycles). There were two domains including *Core emotional symptoms and Secondary symptoms*. A PMDD cycle was identified if participants had at least five symptoms, of which at least one was a core symptom (including Depression, Anxiety, Mood lability, Anger, or Lack of Interest). If the patient had at least two PMDD cycles, a confirmed PMDD diagnosis would be made. All patients screened positive for PMDD using C-PASS were referred to psychiatrists at the Psychological Department of HueUMP.

Furthermore, a PMS cycle was identified based on ACOG criteria [11, 36, 38] that required at least one of the positive emotional symptoms including Depression, Angry, Irritability, Anxiety, Confusion or Social withdrawal and a positive physical symptom. A patient having at least two PMS cycles was diagnosed with PMS.

To assess physical activities, we used IPAQ-SF [39, 40], which was designed to collect information from individuals aged 15–69 years. The Vietnamese version of PHQ-9 was used to screen for depressive symptoms in our study population, which is well described in other countries [41, 42] and Vietnam [32, 40, 43].

## Translation and pilot study

The original English versions of PSST and DRSP were translated first into Vietnamese by two psychiatrists and one gynecologist with internationally recognized English certificates and more than ten years of clinical experience. This translation committee deliberated on the differences in translations and finalized the first Vietnamese version. An independent psychologist, who had a medical English translation degree, back-translated the first version into English. Finally, both English versions were compared by other two specialists (one psychiatrist and one gynecologist) to ensure that the translation was appropriate. The second Vietnamese version were compiled by the committee (including all specialists) and piloted on 30 individuals aged 18 to 45. The participants were asked about their opinions regarding

readability, semantics, understanding, and cultural adaptability. The final versions of these questionnaires were brought into use after reviewing and editing based on the results of the pilot study and the experts' consensus (S1 File in S1 Data).

## Data analysis

**PSST Validation.** Stata 15.1 was used to clean data and calculate the prevalence of PMS and PMDD [44, 45]. The Chi-squared, Fisher exact, Wilcoxson, and Kruskal–Wallis tests were then used to compare the difference among PMS/PMDD and non-PMS/PMDD groups. Using the C-PASS diagnosis as the gold standard [11, 16, 35, 38], we calculated sensitivity, specificity, positive predictive value (PPV), and negative predictive value (NPV) of PSST for PMS and PMDD. The test-retest PSST reliability was conducted at the end of the follow-up phase. We also computed the Kappa coefficient of agreement on classification (into PMS/PMDD and non-PMS/PMDD group).

**Exploratory factor analysis.** Exploratory factor analysis (EFA) was performed to explore the possible latent variables that underlie the question items. We first ran a preliminary model using principal component analysis, then created a scree plot [46] and ran parallel analysis [47] to determine the number of factors to retain. The decision about the number of factors was made based on the following a priori criteria: (a) eigenvalues >1, (b) total variance explained >80%, (c) eigenvalues greater than or equal to the eigenvalue at the "elbow" in the scree plot, and (d) observed eigenvalues greater than the eigenvalue of the same component calculated in parallel analysis.

The EFA model was then estimated using iterated principal factors with the number of factors determined above. Orthogonal rotation was applied to the estimated model to maximize the loading of the items on each factor. Items with uniqueness >0.5, highest loading on a factor <0.4, or high loadings (≥0.4) on >1 factor would be dropped. We also dropped factors that have <3 high-loading (loading ≥0.4) items. The process was iterated until no more items would be dropped from the model. The final model was examined for theoretical meaningfulness.

We would report the result of Bartlett's test of sphericity [48] and the Kaiser-Meyer-Olkin (KMO) measure of sampling adequacy [49] to confirm the usefulness of EFA. We also report the internal consistency of the factors (using Cronbach's alpha).

**Multivariable regression model.** To identify factors associated with PMS/PMDD, a literature review was conducted. From this review, we developed a causal diagram (DAG—Directed acyclic graph) to demonstrate the potential relationship between PMS/PMDD and other factors [50–52]. Many authors stated that the causal inference approaches, such as using DAGs, could deliver more valid estimates than other traditional statistical approaches (e.g, backward selection and forward selection) [50–52].

With the high prevalence of PMS/PMDD among our study population, logistic regression would overestimate the association of independent variables with this binary outcome [53, 54]. Directly estimating Prevalence Ratios (PRs) based on log-binomial regression models might be preferable in this case, however, it is often fails to converge [55]. Zou et al. and Barros et al. showed that using modified Poisson regression model (Poisson regression with a robust error variance) with binary outcome data would be fixed to calculate PRs (or relative risk with cohort study) [53, 54]. Chen et al. also confirmed that log-binomial and modified Poisson regression models produced similar results [56]. Therefore, we calculated PRs using the modified Poisson regression model to assess the association of potential risk factors with PMS/PMDD [44, 53, 54]. Two separate regression models (based on PSST and C-PASS diagnosis) were computed to compare the differences of association factors.

## Results

### Characteristics of participants

The flow of participant recruitment and retention was presented in Fig 1. Among 447 female students who registered to participate in our study, 428 students were eligible and completed the baseline assessment. We excluded 19 patients because of having irregular menstruation (12 students) and a history of endocrine conditions within the last six months (7 students). After the follow-up phase, 302 participants (70.56%) who completed reporting symptoms of at least two consecutive menstrual cycles were included in the data analysis.

Table 1 showed the baseline characteristics of participants. The median age at the study entry was 22.55 years old and age at menarche was 14 years old. About half of the participants were in the general medicine training program. Most of subjects identified as Kinh people, had a positive Rh blood type, and were of normal BMI. About 8.3% (i.e., 1 out of every 12 students) were overweight or obese. While the proportion of consuming alcohol more than once per month was only 9.3%, more than half of study participants reported consuming caffeine more than once per month. About half of the participants appeared to be physically active with moderate or vigorous activity level reported in the past 7 days. No participant reported that they had ever smoked. We also checked the difference in characteristics of 302 participants who completed the study versus 126 lost-to-follow-up participants (S1 Table in S1 Data). The participant group who remained in the study were about one year younger, had slightly lower BMI, and had more PMS/PMDD cases than participants who did not stay in the study.

### Validity of Vietnamese PSST

A comparison was made to estimate the performance of PSST at baseline against the C-PASS (based on the DSM-5 criteria for PMS/PMDD). According to the C-PASS, 35 students (11.6%; 95%CI: 8.2–15.7%) met the diagnosis of PMS (31 students) or PMDD (4 students). Four patients diagnosed with PMDD based on C-PASS was referred to psychiatrist and confirmed to have PMDD. The number of participants with or without PMS/PMDD based on the diagnosis of PSST and CPASS is presented in S2a-S2c Table in S1 Data.

Table 2 indicated fairly good diagnostic accuracy of PSST at baseline compared to C-PASS as the gold standard. PSST at baseline performed quite well in identifying students with PMS/PMDD; with sensitivity of 80.0 (95%CI: 63.1–91.6%) and specificity of 76.8 (95%CI: 71.2–81.7%). However, the diagnostic value of the second PSST at the end of the study compared with C-PASS was much lower (sensitivity: 56.3% (95%CI: 37.7–73.6%); specificity: 78.7% (95% CI: 73.0–83.7%) (S3 Table in S1 Data).

S1 Fig in S1 Data showed the change of mean of the score of 24 symptoms reported via DRSP over two menstrual cycles. During 7 days before menstruation, the participants with PMS/PMDD reported having all symptoms at a noticeably higher severity level compared to counterparts without these conditions. During 10 days after the first menstrual day, the mean score of all symptoms of participants with PMS/PMDD dropped dramatically but still higher than that of participants without PMS/PMDD. The proportions of moderate or severe symptoms detected based on PSST at baseline and end of study are depicted in S2 Fig in S1 Data. Overall, the PMS/PMDD group reported higher severity of symptoms. In the PMS/PMDD group, most symptoms were reported in 10–20% of the participants, with the most prominent symptoms being reported as anger/irritability, anxiety/tension, difficulty concentrating, fatigue/lack of energy, and physical symptoms such as breast tenderness, headaches, joint/muscle pain, bloating, weight gain.

**Table 1. Baseline characteristics of study participants.**

| Characteristics | Total (n = 302) |
|---|---|
| **Age (Years), mean (sd)** | 23.24 (3.58) |
| **Ethnic group, n (%)** | |
| Kinh | 290 (96.0) |
| Others | 12 (4.0) |
| **Medical specialty, n (%)** | |
| General medicine | 163 (54.9) |
| Preventive medicine | 49 (16.5) |
| Traditional medicine | 21 (7.1) |
| Pharmacy | 26 (8.8) |
| Others[a] | 38 (12.8) |
| **Age of menarche (Years), median (IQR)** | 14.00 (13.00; 14.00) |
| **Number of menstruation days (Days), median (IQR)** | 5.00 (4.00; 5.00) |
| **Number of menstrual cycle days (Days), median (IQR)** | 30.00 (28.00; 31.00) |
| **Menstrual blood volumes[b] (ml), median (IQR)** | 68.50 (44.50; 103.25) |
| **BMI (kg/m$^2$), mean (sd)** | 19.94 (2.16) |
| **BMI classification[c], n (%)** | |
| Underweight | 79 (26.2) |
| Normal | 198 (65.6) |
| Overweight & Obese | 25 (8.3) |
| **ABO blood type[d], n (%)** | |
| A | 51 (16.9) |
| B | 88 (29.1) |
| AB | 19 (6.3) |
| O | 128 (42.4) |
| Unknown | 16 (5.3) |
| **Rh blood type[d], n (%)** | |
| Positive | 269 (89.1) |
| Negative | 11 (3.6) |
| Unknown | 22 (7.3) |
| **Alcohol consumption of the last 12 months, n (%)** | |
| No | 48 (15.9) |
| Once per month or less | 226 (74.8) |
| More than once per month | 28 (9.3) |
| **Caffeine consumption of the last 12 months, n (%)** | |
| Once a month or less | 158 (52.3) |
| 2–3 times per month to 1–3 times per week | 117 (38.7) |
| From 4 times per week and above | 27 (8.9) |
| **Physical Activity in the last 7 days[e], n (%)** | |
| Low | 145 (48.0) |
| Moderate | 108 (35.8) |
| Vigorous | 49 (16.2) |

[a]Including Dentistry, Public health, Medical technician, and Nursing.

[b]Menstrual blood volumes were estimated via menstrual pictograms (SAP-c version).

[c]BMI was classified according to the Asia-Pacific body mass index classifications: Underweight (<18.5 kg/m2),

Normal (18.5–23 kg/m2), Overweight (23–27.5 kg/m2) and Obese (> 27.5 kg/m2)

[d]Blood types were self-reported.

[e]The physical activity levels were classified according to the International Physical Activity Questionnaire.

**Abbreviation**: BMI—Body measurement index; SD—standard deviation; IQR—interquartile range.

**Table 2. Sensitivity and specificity of PSST (for PMS & PMDD) compared with C-PASS.**

| Indicators | Estimate | 95% Confidence interval |
|---|---|---|
| Sensitivity (%) | 80.0 | 63.1–91.6 |
| Specificity (%) | 76.8 | 71.2–81.7 |
| Likelihood ratio (+) | 3.45 | 2.62–4.53 |
| Likelihood ratio (-) | 0.26 | 0.13–0.51 |
| Positive predictive value (%) | 31.1 | 21.8–41.7 |
| Negative predictive value (%) | 96.7 | 93.3–98.7 |
| Kappa (PSST vs CPASS) | 0.337 | 0.226–0.449 |
| Percent agreement (%) | 77.15 | N/A |

**Abbreviation**: *PSST (Premenstrual Syndrome Screening Tools); C-PASS (Carolina Premenstrual Assessment Scoring System); PMS (Premenstrual syndrome); PMDD (Premenstrual dysphoric disorders).*

## Exploratory factor analysis

The preliminary analysis of EFA suggested that two or three factors were adequate (S3 Fig in S1 Data). After refining our models, we determined that the two-factor model including six items (B5, B6, B7, B16, B17, and B18) was appropriate and parsimonious. S4 Table in S1 Data presents the factor loadings after orthogonal rotation. The factor loading plot shows two groups of items with high loadings on one of the two factors (S3 Fig in S1 Data). Based on the content of the items, we determined that these two factors represent decreased interests in usual activities (B5, B6, and B7) and problems with relationships (B16, B17, and B18). The factors had good internal consistency (Cronbach's alphas 0.80 and 0.84, respectively). The Barlett's test of sphericity p-value was <0.0001 and the KMO statistic was 0.79, suggesting the EFA model was adequate.

## Associated factors

Table 3 compared the characteristics of participants by PMS/PMDD status according to the C-PASS diagnosis. Interestingly, the proportion of having a Rh-negative blood type in the PMS/PMDD group (4 out of 35) is significantly higher than that of the non-PMS/PMDD group (7 out of 267) (p-value = 0.013). The students with PMS/PMDD also had a lower proportion of late menarche ($\geq$ 15 years old), reported consuming caffeine more frequently and being more physically active. There was no significant difference between the PMS/PMDD group and non-PMS/PMDD group in terms of ABO blood types, BMI, obstetrical history, family history of psychological disorders, alcohol consumption, physical activity, and depression level. When using PSST at baseline as the diagnosis test, only caffeine consumption and depression levels were significantly different between the two student groups (S5 Table in S1 Data).

We used a multivariable modified Poisson regression model to explore the association between participants' characteristics and PMS/PMDD diagnosed by C-PASS (Table 4). We found that age at menarche (PR = 0.77, 95%CI: 0.63–0.96), having a negative Rh blood type (PR = 4.43, 95%CI: 1.95 to 10.08), being moderately depressed or higher (PR = 2.81, 95%CI: 1.24 to 6.36), and consuming caffeine more than three times per week were statistically associated with having PMS/PMDD after adjusting for other variables. In the multivariable modified Poisson regression model with PMS/PMDD as the outcome diagnosed by PSST at baseline, only consuming caffeine more than three times per week was associated with an increased

**Table 3. Characteristics of participants by PMS/PMDD status according to the C-PASS diagnosis.**

| | No PMS & PMDD | PMS or PMDD | P-value |
|---|---|---|---|
| **n (%)** | **267 (88.4)** | **35 (11.6)** | |
| **Biological and physical measurement** | | | |
| **Age (Years), mean (sd)** | 23.30 (3.70) | 22.75 (2.54) | 0.396[1] |
| **ABO blood type[a], n (%)** | | | |
| A | 46 (90.2) | 5 (9.8) | |
| B | 76 (86.4) | 12 (13.6) | |
| AB | 16 (84.2) | 3 (15.8) | |
| O | 113 (88.3) | 15 (11.7) | |
| Unknown | 16 (100.0) | 0 (0.0) | 0.588[2] |
| **Rh blood type[a], n (%)** | | | |
| Positive | 238 (88.5) | 31 (11.5) | |
| Negative | 7 (63.6) | 4 (36.4) | |
| Unknown | 22 (100.0) | 0 (0.0) | 0.013[2] |
| **Study year, n (%)** | | | |
| Preclinic ($< = 3$ years) | 128 (89.5) | 15 (10.5) | |
| Clinic ($> 3$ years) | 139 (87.4) | 20 (12.6) | 0.594[2] |
| **BMI (kg/m2), mean (sd)** | 19.87 (2.17) | 20.52 (2.03) | 0.091[1] |
| **BMI classification[b], n (%)** | | | |
| Underweight | 73 (92.4) | 6 (7.6) | |
| Normal | 173 (87.4) | 25 (12.6) | |
| Overweight & Obese | 21 (84.0) | 4 (16.0) | 0.357[2] |
| **Menstrual status** | | | |
| **Age of menarche (Years), median (IQR)** | 14.00 (13.00; 15.00) | 13.00 (13.00; 14.00) | 0.049[3] |
| **Late menarche ($> = 15$ years old), n (%)** | 70 (95.9) | 3 (4.1) | 0.021[2] |
| **Menstrual days (Days), median (IQR)** | 5.00 (4.00; 5.00) | 5.00 (4.00; 5.00) | 0.706[3] |
| **Cycle days (Days), median (IQR)** | 30.00 (28.00; 31.00) | 30.00 (29.00; 31.00) | 0.373[3] |
| **Menstrual blood volumes (ml), median (IQR)** | 68.50 (44.25; 104.50) | 66.50 (41.50; 91.50) | 0.888[3] |
| **Obstetric history** | | | |
| **Had $> = 1$ pregnancy, n (%)** | 20 (95.2) | 1 (4.8) | 0.487[2] |
| **History of C-section, n (%)** | 7 (87.5) | 1 (12.5) | 1.000[2] |
| **History of Term births, n (%)** | 18 (94.7) | 1 (5.3) | 0.709[2] |
| **History of Preterm births, n (%)** | 3 (100.0) | 0 (0.0) | 1.000[2] |
| **History of Abortions, n (%)** | 3 (75.0) | 1 (25.0) | 0.391[2] |
| **Family history** | | | |
| **Psychological disorders in 1st degree relatives[c], n (%)** | 5 (83.3) | 1 (16.7) | 0.526[2] |
| **Lifestyle** | | | |
| **Alcohol consumption in the last 12 months, n (%)** | | | |
| No | 43 (89.6) | 5 (10.4) | |
| Once per month or less | 200 (88.5) | 26 (11.5) | |
| More than once per month | 24 (85.7) | 4 (14.3) | 0.907[2] |
| **Caffeine consumption in the last 12 months, n (%)** | | | |
| Once a month or less | 141 (89.2) | 17 (10.8) | |
| 2–3 time per month to 1–3 times per week | 107 (91.5) | 10 (8.5) | |
| From 4 times per week and above | 19 (70.4) | 8 (29.6) | 0.014[2] |
| **Physical Activity in the last 7 days[d], n (%)** | | | |
| Low | 133 (91.7) | 12 (8.3) | |
| Moderate | 95 (88.0) | 13 (12.0) | |
| Vigorous | 39 (79.6) | 10 (20.4) | 0.075[2] |

(*Continued*)

**Table 3.** (Continued)

| | No PMS & PMDD | PMS or PMDD | P-value |
|---|---|---|---|
| **STRESS** | | | |
| **Depression level based on PHQ-9, n (%)** | | | |
| No or Minimal depression | 100 (90.1) | 11 (9.9) | |
| Mild depression | 131 (89.1) | 16 (10.9) | |
| Moderate depression | 36 (81.8) | 8 (18.2) | |
| Severe depression | 0 (0.0) | 0 (0.0) | 0.342[2] |

[a] Blood type was self-reported.

[b]BMI was classified according to the Asia-Pacific body mass index classifications: Underweight ($<18.5$ kg/m$^2$), Normal (18.5–23 kg/m$^2$), Overweight (23–27.5 kg/m$^2$) and Obese ($> 27.5$ kg/m$^2$)

[c]In the non-PMS/PMDD group, the family history of psychological disorders included depression (n = 2), Anxiety (n = 2), Schizophrenia (n = 1), and bipolar disorder (n = 1), while that of the PMS/PMDD group included depression (n = 1). The family history was self-reported and referred to their first-degree relatives.

[d]Physical activity was assessed by the International Physical Activity Questionnaire (IPAQ-SF)

**Statistical test**:

[1]Kruskal–Wallis tests,

[2]Fisher exact and

[3]Wilcoxson.

**Abbreviation**: *BMI—Body mass index; PSST—Premenstrual Syndrome Screening Tools; C-PASS—Carolina Premenstrual Assessment Scoring System; PMS—Premenstrual syndrome; PMDD—Premenstrual dysphoric disorders; PHQ-9—Patient health questionare 9.*

prevalence of PMS/PMDD compared to participants consuming once per month or less (S5 Table in S1 Data).

## Discussions

According to the C-PASS, 11.6% of the study sample (95%CI: 8.2–15.7%) met the diagnosis of PMS or PMDD. Based on C-PASS as the gold standard, the PSST demonstrated good validity in the role of the screening test with high sensitivity (80.0%; 95% CI 63.1–91.6%), specificity (76.8%; 95% CI 71.2–81.7%), extremely high negative predictive value—NPV (97.2%; 95% CI 93.3–98.7%), but low positive predictive value—PPV (31.1%, 95% CI 21.8–41.7%) (Table 5). We found that age at menarche (PR = 0.77, 95%CI: 0.63–0.96), having a negative Rh blood type (PR = 4.43, 95%CI: 1.95 to 10.08), being moderately depressed or higher (PR = 2.81, 95% CI: 1.24 to 6.36), and consuming caffeine more than three times per week were statistically associated with having PMS/PMDD after adjusting for other variables.

According to the multiple regression analyses, there were four significant associated factors of PMS/PMDD including late menarche, a negative Rhesus blood type, caffeine consumption, and depression (screened by PHQ-9). We found that higher age of menarche was a protective factor for PMS/PMDD (PR = 0.77, 95% CI 0.63–0.96). Our result was consistent with the study of Donghao Lu et al. that reported that women with late menarche had a lower risk of PMS/PMDD (OR 0.73, 95% CI 0.59 to 0.91) [57]. The mechanism by which late menarche affects PMS and PMDD remained unclear. However, there has been evidence that the elevation and cyclicity of hormones during puberty regulates the sensitivity of the neuroendocrine system, which could simulate PMS/PMDD in some individuals [58, 59]. Therefore, it was possible that early or late exposure to the elevation and cyclicity of hormones might result in a different risk of PMS/PMDD.

**Table 4. Multivariable Poisson regression model of PMS/PMDD diagnosed by CPASS* and PSST.**

| PMS/PMDD diagnosis based on C-PASS | CPASS (n = 279) | | | | PSST (n = 279) | | | |
|---|---|---|---|---|---|---|---|---|
| | PR | 95% CI | | P—value | PR | 95% CI | | P—value |
| **Age of menarche** | **0.777** | **0.631** | **0.956** | **0.017** | 0.911 | 0.789 | 1.051 | 0.200 |
| **Number of menstrual days** | 0.958 | 0.835 | 1.101 | 0.548 | 1.003 | 0.947 | 1.062 | 0.924 |
| **Number of cycle days** | 1.086 | 0.985 | 1.196 | 0.096 | 0.986 | 0.920 | 1.058 | 0.699 |
| **Menstrual blood volume** | 0.997 | 0.991 | 1.004 | 0.416 | 0.997 | 0.993 | 1.001 | 0.128 |
| **Rh blood type** | | | | | | | | |
| Positive *(Ref)* | 1.000 | | | | 1.000 | | | |
| Negative | **4.428** | **1.946** | **10.076** | **<0.001** | 0.884 | 0.264 | 2.956 | 0.841 |
| **ABO blood type** | | | | | | | | |
| A *(Ref)* | 1.000 | | | | 1.000 | | | |
| B | 1.570 | 0.619 | 3.984 | 0.342 | 1.447 | 0.815 | 2.568 | 0.207 |
| AB | 2.704 | 0.699 | 1.046 | 0.149 | 1.500 | 0.598 | 3.759 | 0.387 |
| O | 1.625 | 0.632 | 4.180 | 0.313 | 1.403 | 0.794 | 2.479 | 0.243 |
| **BMI classification** | | | | | | | | |
| Normal *(Ref)* | 1.000 | | | | 1.000 | | | |
| Underweight | 0.587 | 0.245 | 1.405 | 0.232 | 1.503 | 0.941 | 2.400 | 0.088 |
| Overweight & Obese | 0.983 | 0.368 | 2.623 | 0.973 | 0.924 | 0.437 | 1.957 | 0.837 |
| **Depression level based on PHQ-9** | | | | | | | | |
| No or Minimal depression *(Ref)* | 1.000 | | | | 1.000 | | | |
| Mild depression | 1.393 | 0.686 | 2.831 | 0.359 | 1.340 | 0.910 | 1.973 | 0.139 |
| Moderate depression | **2.807** | **1.239** | **6.357** | **0.013** | 1.386 | 0.850 | 2.259 | 0.191 |
| **Alcohol consumption in the last 12 months** | | | | | | | | |
| No *(Ref)* | 1.000 | | | | 1.000 | | | |
| Once per month or less | 0.733 | 0.282 | 1.903 | 0.523 | 0.937 | 0.519 | 1.691 | 0.829 |
| More than once per month | 0.968 | 0.309 | 3.030 | 0.955 | 1.307 | 0.635 | 2.689 | 0.468 |
| **Caffeine consumption in the last 12 months** | | | | | | | | |
| Once a month or less *(Ref)* | 1.000 | | | | 1.000 | | | |
| 2–3 times per month to 1–3 time per week | 0.901 | 0.443 | 1.829 | 0.772 | 1.304 | 0.897 | 1.895 | 0.164 |
| From 4 times per week and above | **2.858** | **1.241** | **6.578** | **0.014** | **2.186** | **1.363** | **3.505** | **0.001** |
| **Physical Activity (IPAQ)** | | | | | | | | |
| Low *(Ref)* | 1.000 | | | | 1.000 | | | |
| Moderate | 1.701 | 0.813 | 3.556 | 0.158 | 1.340 | 0.910 | 1.973 | 0.139 |
| Vigorous | 1.592 | 0.701 | 3.616 | 0.267 | 1.386 | 0.850 | 2.259 | 0.191 |

*We excluded 23 participants who did not report their blood type (ABO blood type and/or Rh blood type) and performed analysis on 279 samples.

**Abbreviation**: *BMI—Body mass index; PSST—Premenstrual Syndrome Screening Tools; C-PASS—Carolina Premenstrual Assessment Scoring System; PMS—Premenstrual syndrome; PMDD—Premenstrual dysphoric disorders; PHQ-9—Patient health questionare 9; IPAQ—International Physical Activity Questionnaire.*

In our study, the Poisson regression analysis did not show a significant association regarding ABO blood type and PMS/PMDD; however, a negative Rh blood type was the most critical factor associated with an increased risk of PMS/PMDD (PR = 4.43, 95% CI: 1.95–10.08). PurohitKanhu Charan et al. showed an association between ABO blood groups with PMS [60]. They also reported that those with an O blood type had a significant proportion of headache and abdominal pain, while the AB group did not experience PMS symptoms [60]. Additionally, PMS had been reported to be higher in identical twins [61] and those who have suffered a mutation of the follicle-stimulating hormone receptor (FSH-R) gene [62]. The mutation may lead to amenorrhea, infertility, or premature ovarian failure indicating that genetic

**Table 5. Summary of studies on the validity of Premenstrual Syndrome Screening Tool.**

| Author | Country | Subjects | Sample | Diagnostic criteria | PMS | PMDD | Sen | Spe | PPV | NPV |
|---|---|---|---|---|---|---|---|---|---|---|
| Raval, 2014 | India | Female students | 529 | SCID-PMDD/ DSM-IV-TR | 14,7% | 3,7% | 90,9% | 58,01% | 28,9% | 97,0% |
| Mirghafourv, 2015 | Iran | Female students | 230 | DRSP/ DSM-5 | 83.9% | N/A | 66,3% | 85,6% | 96,2% | 33,0% |
| Mahfoud, 2019 | Arab | Women | 194 | MINI-PLUS 6 / DSM-5 | 37% | 15% | 81,5% | 61,6% | 85.2% | 58.3% |
| Henz, 2018 | Brazil | Women | 127 | DRSP / DSM-5 | 74,8% | 3,9% | 79% | 33% | 81,4% | 30,0% |
| **Our study, 2021** | Vietnam | Medical female-students | 302 | DRSP/DSM-5 | 10,3% | 1,3% | 80,0% | 76,8% | 33.0% | 96.7% |

**Abbreviation**: *PMS—Premenstrual syndrome; PMDD—Premenstrual dysphoric disorders; Sen—Sensitivity; Spe—Specificity; PPV—Positive predictive value; NPV—Negative predictive value.*

susceptibility is significantly related to PMS. Young Su et al. reported that the ABO blood group gene and TRAF2 gene might cause menstrual disorders (MD), including PMS/PMDD [63]. They found that genotype distribution frequencies of *rs657152* and *rs495828* loci, located upstream of the ABO gene, were significantly higher in the MD group than the control group [63]. It suggested genetic factors related to blood-type might hidden in association with menstrual disorders, which could be more investigated.

Interestingly, we did not find any studies related to Rh blood type and premenstrual disorders. However, several studies demonstrated that Rh phenotype and genotype could significantly affect human psychomotor performance [64–66] and health [67–69]. Generally, Rh-negative individuals might suffer worser health, a low score in some performance tests, and a higher sensitivity to varied adverse environmental exposures [69]. Rh-positive blood types were found to possibly have good-quality health and performance than others [69], suggesting that PMS/PMDD could be conditions susceptible to Rh-negative subjects.

Consistent with previous studies, caffeine consumption (more than three times per week) was found to be a major factor associated with PMS/PMDD (PR = 2.86, 95% CI: 1.24–6.58) in our study [70–73]. The essential biological mechanisms of PMS/PMDD were the decrease of serotonin levels during the luteal phase and the significant decline of blood platelet serotonin (5-HT) levels before menstruation [74, 75]. On the other hand, a previous study reported that caffeine could reduce serotonin synthesis by inhibiting tryptophan hydroxylase and blocking adenosine α1 and α2 receptors within the central nervous system [76, 77]. The condition could exacerbate the serotonin decrease and worsen PMS/PMDD symptoms.

We also found that depression level showed a positive association with PMS/PMDD (PR = 2.81, 95% CI: 1.24–6.36), which was also reported in several previous findings [78–81]. Depression was also considered a symptom of PMS/PMDD, but it only became more obvious or severe several days before menstruation [10, 35, 82]. Monitoring the severity of symptoms during the menstrual cycle allowed identification of the variation of symptoms before and after menstruation and distinguishing them from persistent depression [10, 11]. There could be complicated interactions between PMS/PMDD and depression, which has been previously reported. However, the results could be skewed due to reverse causation or unknown confounding; the investigation period might overlap the luteal phase which expressed high severity of depression; or the medical student population having a higher likelihood of depression [83].

We also found a difference between the two models of multivariable regression analysis based on diagnosis by CPASS and PSST. Based on PSST diagnosis, only caffeine consumption was associated with PMS/PMDD (PR = 2.19; 95% CI: 1.37–3.51). This result showed that, although PSST has high sensitivity and specificity in diagnosing PMS/PMDD, the misclassification of PSST compared to C-PASS attenuated the power to detect significant risk factors with PMS/PMDD in this study.

Focusing on the many strengths of this study, firstly, it was the first study in Vietnam and one of the very few in Southeast Asia using follow-up data from at least two menstrual cycles using DRSP, which was the gold standard in the PMS/PMDD diagnosis. Second, the diagnosis protocol was rigorous as participants who screened positive were later checked by a psychiatrist to obtain final diagnosis. However, this study has a few limitations. First, the sample size was small and only restricted to female medical students which is not representative of Vietnamese population aged 18–45; therefore, the generalizability of the study results to all Vietnamese women is weak. Second, the cross-sectional study design does not allow us to identify the causal relationship of the factors and the outcome, which makes the results prone to reverse causation. Third, our exclusion criteria on having any history of mental health disorders were self-reported, which might be underdiagnosed or underreported. Finally, there are still residual and unmeasured confounding that we cannot control for in the models.

## Conclusion

PMS/ PMDD are relatively common disorders in our study. The prominent risk factors for PMS and PMDD were negative Rhesus blood type, menarche age, caffeine consumption, and self-reported depression. The Vietnamese PSST was shown as an effective screening tool for PMS/PMDD with strong validity compared to the gold standard C-PASS. Monitoring the burden of these conditions over time and determining both modifiable factors that can alleviate or exacerbate the symptoms would be important for managing health of not only women but also their social connections.

## Supporting information

**S1 Data.**
(ZIP)

## Acknowledgments

We warmly thank our colleagues at Obstetrics and Gynaecology student's club (OGC), and medical students at HueUMP for assisting our group. We would like to recognize the academic support of Dr. Nguyen Le Hung Linh, Dr. Tran Thi My Duyen and Dr. Tran Hoang Nhat Anh right from the start of this project. We also thank Dr. Tran Nhu Minh Hang, Dr. Nguyen Huu Cat, and Dr. Ho Dung for their assistance with independent translating and comments on the Vietnamese PSST version. Also, we thank Nilendra Nair for providing language editing of the manuscript.

## Author Contributions

**Conceptualization:** Vy Dinh Trieu Ngo, Linh Phuong Bui, Long Bao Hoang, Linh Manh Tran.

**Data curation:** Vy Dinh Trieu Ngo, Linh Phuong Bui, Linh Manh Tran.

**Formal analysis:** Vy Dinh Trieu Ngo, Linh Phuong Bui, Long Bao Hoang, Linh Manh Tran, Tung Thanh Pham.

**Funding acquisition:** Vy Dinh Trieu Ngo, Linh Manh Tran.

**Investigation:** Vy Dinh Trieu Ngo, Linh Phuong Bui, Long Bao Hoang, My Thi Tra Tran, Huy Vu Quoc Nguyen, Linh Manh Tran, Tung Thanh Pham.

**Methodology:** Vy Dinh Trieu Ngo, Linh Phuong Bui, Long Bao Hoang, My Thi Tra Tran, Huy Vu Quoc Nguyen, Linh Manh Tran, Tung Thanh Pham.

**Project administration:** Vy Dinh Trieu Ngo, Linh Phuong Bui, Long Bao Hoang, Huy Vu Quoc Nguyen, Linh Manh Tran, Tung Thanh Pham.

**Resources:** Vy Dinh Trieu Ngo, Linh Phuong Bui, Long Bao Hoang, My Thi Tra Tran, Huy Vu Quoc Nguyen, Linh Manh Tran, Tung Thanh Pham.

**Software:** Vy Dinh Trieu Ngo, Linh Phuong Bui, Long Bao Hoang, Tung Thanh Pham.

**Supervision:** Vy Dinh Trieu Ngo, Linh Phuong Bui, Long Bao Hoang, My Thi Tra Tran, Huy Vu Quoc Nguyen, Linh Manh Tran, Tung Thanh Pham.

**Validation:** Vy Dinh Trieu Ngo, Linh Phuong Bui, Long Bao Hoang, My Thi Tra Tran, Huy Vu Quoc Nguyen, Linh Manh Tran, Tung Thanh Pham.

**Visualization:** Vy Dinh Trieu Ngo, Linh Phuong Bui, Long Bao Hoang, My Thi Tra Tran, Huy Vu Quoc Nguyen, Linh Manh Tran, Tung Thanh Pham.

**Writing – original draft:** Vy Dinh Trieu Ngo, Linh Phuong Bui, My Thi Tra Tran, Huy Vu Quoc Nguyen, Linh Manh Tran, Tung Thanh Pham.

**Writing – review & editing:** Vy Dinh Trieu Ngo, Linh Phuong Bui, Long Bao Hoang, My Thi Tra Tran, Huy Vu Quoc Nguyen, Linh Manh Tran.

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
