## [Decision Letter · Decision Letter 0]

13 Oct 2022

PONE-D-22-24425Associated Factors with Premenstrual Syndrome and Premenstrual Dysphoric Disorder Among Female Medical Students.PLOS ONE

Dear Dr. Ngo Dinh Trieu,

Thank you for submitting your manuscript to PLOS ONE. After careful consideration, we feel that it has merit but does not fully meet PLOS ONE’s publication criteria as it currently stands. Therefore, we invite you to submit a revised version of the manuscript that addresses the points raised during the review process.

Please take into consideration the reviewers comments regarding:

1- Expressing the correct aim of the study ( wich cannot be to find the risk factors in the population , but rather within a sample od medical students aged 18-45

2-Stating in a stronger manner the main limitations ( cross sectional design and lck of capacity for causal inference, and only capability of stating hypothesis AND weak external validity for two reasons : 1- medical students aged 18-45 are not representative of women 18-45 in Vietnam 2- age distribution within the studied population will probably not match the age distribution of the population. (was this the only problem, it could have been age- standardized)

We look forward to receiving your revised manuscript.

Kind regards,

Ignacio Garitano Gutierrez, MD, MSc

Academic Editor

PLOS ONE

Journal Requirements:

"FUNDING

The study had been supported by the Research Advancement Consortium in Health and Hue University of Medicine and Pharmacy, Vietnam (https://www.reach.edu.vn/pmspmdd.html). "

4. Thank you for stating the following in the Funding Section of your manuscript: 

"The study had been supported by the Research Advancement Consortium in Health and Hue University of Medicine and Pharmacy, Vietnam (https://www.reach.edu.vn/pmspmdd.html).  "

"FUNDING

The study had been supported by the Research Advancement Consortium in Health and Hue University of Medicine and Pharmacy, Vietnam (https://www.reach.edu.vn/pmspmdd.html).  "

"Vy. D. T. Ngo has received a research grant from Research Advancement Consortium in Health (REACH) which Linh P. Bui, Long B. Hoang and Tung T. Pham are founders and members of REACH (a non-profit entity). Linh P. Bui, Long B. Hoang and Tung T. Pham did not receive any payment or compensation from this position at REACH. They provided consultancy on study design, data collection and analysis, and preparation of the manuscript. However, REACH had no role in the decision to publish this study, and this final decision belongs to the funded research team."

Additional Editor Comments:

Please fill in the ethics statement box of the questionnaire with the approval by the Institutional Review Board of Hue University of Medicine and Pharmacy with the registration number: H2019/003.

I would include the kind of study within the title.

Regarding the aim of the study in the abstract, I would rephrase:

"The study aimed at determining potential risk factors associated with Premenstrual

Syndrome (PMS) and Premenstrual Dysphoric Disorder (PMDD)."

As a cross sectional study I would rater talk about associated factors and potential risk factors. Do not try to make them look causal.

The main limitation apart from the cross sectional character of the study is the weak external validity, meaning that the studied population is probably not representative of the Vietnamese female population aged 18-45 years.

This should be taken into account and of course displayed as a limitation in a more strong manner. ( The authors say "limits the generalizability", when they should straightly say, "results are not generalizable to the vietnamese female population aged 18-45."

Line 58, what geographic area are the authors talking about?

Line 85 please specify that the study aim is targeted to a vietnamese female medical students sample.

Figure 1

Among 447 screened women there was not a single case of endometriosis ( according to the literature around 10% of women suffer it ) please discuss as a limitation.

Reviewers' comments:

Reviewer's Responses to Questions

**Comments to the Author**

1. Is the manuscript technically sound, and do the data support the conclusions?

Reviewer #1: Partly

Reviewer #2: Partly

2. Has the statistical analysis been performed appropriately and rigorously? 

Reviewer #1: Yes

Reviewer #2: Yes

3. Have the authors made all data underlying the findings in their manuscript fully available?

Reviewer #1: Yes

Reviewer #2: Yes

4. Is the manuscript presented in an intelligible fashion and written in standard English?

Reviewer #1: Yes

Reviewer #2: Yes

5. Review Comments to the Author

Reviewer #1: 1. Title:

Number of appropriate words and is related to the objectives of the text and the development of the text.

The main objectives of the study are described.

2. Abstract:

It describes how the study was carried out, including the methodology, without delving into the methodological details.

It is recommended that abbreviations not be included in the abstract. Abbreviations appear twice, in the abstract and in the text, remove them from the abstract.

The most important results were summarized with their interpretation.

The abstract does not exceed 300 words.

3. Introduction:

It is recommended to write more than the types of treatment used for each disease.

4. Methods:

Place the name of figure 1 and briefly explain the main losses of participants during the writing in the text.

Inclusion and elimination criteria are missing in the literature review. From what year to what year was the bibliography used, what types of text were used and in what search engines did they find them.

5. Results:

The most important results are highlighted in the text, the tables and figures coincide with what is cited in the text.

6. Conclusion:

The conclusion does not appear in the text.

7. References:

The references are correctly cited in the Vancouver format and in the correct order.

Excellent study, with great impact on women to know the risk factors associated with Premenstrual Syndrome and Premenstrual Dysphoric Disorder, the objectives proposed at the beginning of the study were completed, with adequate methodology that explained step by step how they approached the study. The results were adequately expressed and an excellent review of the literature was carried out. I recommend this article for publication with the minor revision.

Reviewer #2: The study aims to determine possible risk factors for PMS/PMDD in a Vietnamese specific population of students based on previous research about the topic in general population. The methodology is thoroughly described, with several steps, and focused on statistical analysis of the factors included in the study and in the validation of screening and diagnostic tools. Diagnostic process is adequately proposed.

Evaluating it at a whole, the statistics and description of results are thorough, concrete and described. The diagnostic process is clear with some specifications. What might be proposed to review could be the references to the population of the study and the selection of the sample.

Specifying it into notes:

1. Introduction

- The motivation of the study is adequately presented at the introduction. However, the definition and specific risk factors already studied are not clearly addressed and a thorough description could be more accurate as is the reason for the study. PMS/PMDD are sometimes referred as interchangeable, and it is necessary to specify whether evaluation tools are used for diagnosis or screening, since the guidelines set some recommendations about it. I would suggest that references from the International Society of Premenstrual Disorders (O’Brien et al) about consensus in diagnostic criteria could be included.

2. Methods

- The population indicated (women between 18 and 45 years old) might not represent appropriately the population the study is focused on, and so the conclusions might not match with the aim of the study. University students from a specific region in Vietnam could represent the sample and population. Furthermore, prevalence might change, as it has been described higher prevalence among students and women in their twenties.

- Exclusion criteria are described and, as indicated in the algorithm of the study, only few of them make a true exclusion. Some studies refer more specifically to women taking antidepressants, define more concretely diseases or conditions needed to be considered in this setting and evaluate mental disease before consider inclusion as it might be underdiagnosed or difficult to determine once the diagnosis of PMS/PMDD is considered.

- Sample size is calculated based on reported prevalence of PMS (not the described in Vietnamese population -10%-), but not PMDD (much lower prevalence described). Furthermore, the prevalence is established based on diagnostic criteria, with variable match in case of screening tools (PSST), which is the test used for the sample in the study.

- Cases considered from DRSP are referred to a psychiatrist but there is no reference about whether they were confirmed or excluded.

3. Results

- As I observe at Table 1, more than half of the sample (or approximately half) reported consuming caffeine once or less per month.

4. Discussion

- An evaluation of why there is a change of sensitivity of PSST from pre to re-test would be interesting.

- The conclusion that attaches specifically to the aim of the study, is not at the end of the article.

6. PLOS authors have the option to publish the peer review history of their article (what does this mean?). If published, this will include your full peer review and any attached files.

Reviewer #1: No

Reviewer #2: No

---

## [Author Response · Author response to Decision Letter 0]

16 Nov 2022

Dear Editors, 

We would like to express our sincere thanks for the editor and reviewers for your comments on our manuscript. We hope that this revised version of the manuscript will be considered for publication. 

We have modified the paper in response to the extensive and insightful reviewers’ comments. Furthermore, we have rewritten sections of the manuscript and hope that this complies with the reviewers’ remarks. 

We have responded to the comments point by point. Our answers to reviews’ questions below were in blue. All the line numbers mentioned were based on the marked-up copy (the revised manuscript with track changes).

Academic editor review:

Thank you for submitting your manuscript to PLOS ONE. After careful consideration, we feel that it has merit but does not fully meet PLOS ONE’s publication criteria as it currently stands. Therefore, we invite you to submit a revised version of the manuscript that addresses the points raised during the review process.

Please take into consideration the reviewer’s comments regarding:

1- Expressing the correct aim of the study (which cannot be to find the risk factors in the population, but rather within a sample of medical students aged 18-45)

Author’s response: Author’s response: Thank you for your comments. We have updated the aim of the study that “This study aimed to determine potential risk factors associated with PMS/PMDD among the Vietnamese medical student aged 18 – 45 in Hue city via C-PASS or PSST” in line 294.

2-Stating in a stronger manner the main limitations (cross-sectional design and lack of capacity for causal inference, and only capability of stating hypothesis 

Author’s response: Thank you for your great comments. We have further acknowledged the limitations of the study in a stronger manner. At the line 602, we stated that “Second, the cross-sectional study design does not allow us to identify the causal relationship of the factors and the outcome, which makes the results prone to reverse causation”. 

AND weak external validity for two reasons: 1- medical students aged 18-45 are not representative of women 18-45 in Vietnam

Author’s response: Thank you for your great comment. We have additionally stated this limitation of the study that “First, the sample size was small and only restricted to female medical students which is not representative of Vietnamese women aged 18-45; therefore, the generalizability of the study results to all Vietnamese women is weak” in line 600.

 2- age distribution within the studied population will probably not match the age distribution of the population. (was this the only problem, it could have been age-standardized)

Author’s response: Thank you for your recommendation. Because our non-random sample size was small (n=302) and most of the participants in our study were under 24 years old, we believe that age-standardization could not be feasible and meaningful. We have acknowledged the weak external validity of our results in the sentence “First, the sample size was small and only restricted to female medical students which is not representative of Vietnamese women aged 18-45; therefore, the generalizability of the study results to all Vietnamese women is weak” in line 600. 

● A rebuttal letter that responds to each point raised by the academic editor and reviewer(s). You should upload this letter as a separate file labeled 'Response to Reviewers'.

● A marked-up copy of your manuscript that highlights changes made to the original version. You should upload this as a separate file labeled 'Revised Manuscript with Track Changes.

● An unmarked version of your revised paper without tracked changes. You should upload this as a separate file labeled 'Manuscript'.

We look forward to receiving your revised manuscript.

Kind regards,

Ignacio Garitano Gutierrez, MD, MSc

Academic Editor

PLOS ONE

Journal Requirements:

Author’s response: Thank you for your reminder. We have double-checked and tried to make sure the format of manuscript is aligned with PLOS ONE’s style requirements. 

Author’s response: Thank you for your comment. We have added the type of consent in the Methods section. It now reads: “We obtained verbal and written consent from the participants after explaining the relevant information such as study context, objectives, data collection procedure” (please see line 297).

"FUNDING

The study had been supported by the Research Advancement Consortium in Health and Hue University of Medicine and Pharmacy, Vietnam (https://www.reach.edu.vn/pmspmdd.html). "

Author’s response: Thank you for you comment. The statement “The funders had no role in study design, data collection and analysis, decision to publish, or preparation of the manuscript” has added in the cover letter (please see the Cover letter, line 24)

4. Thank you for stating the following in the Funding Section of your manuscript: 

"The study had been supported by the Research Advancement Consortium in Health and Hue University of Medicine and Pharmacy, Vietnam (https://www.reach.edu.vn/pmspmdd.html). "

"FUNDING

The study had been supported by the Research Advancement Consortium in Health and Hue University of Medicine and Pharmacy, Vietnam (https://www.reach.edu.vn/pmspmdd.html). "

Author’s response: Thank you for your reminder. The funding-related text has been removed from the manuscript. We would like to update the Funding statement as “The study had been supported by the Research Advancement Consortium in Health and Hue University of Medicine and Pharmacy, Vietnam (https://www.reach.edu.vn/pmspmdd.html).” The statement also was added in the cover letter at the line 18.

"Vy. D. T. Ngo has received a research grant from Research Advancement Consortium in Health (REACH) which Linh P. Bui, Long B. Hoang and Tung T. Pham are founders and members of REACH (a non-profit entity). Linh P. Bui, Long B. Hoang and Tung T. Pham did not receive any payment or compensation from this position at REACH. They provided consultancy on study design, data collection and analysis, and preparation of the manuscript. However, REACH had no role in the decision to publish this study, and this final decision belongs to the funded research team."

Author’s response: Thank you for your reminder. The statement “This does not alter our adherence to PLOS ONE policies on sharing data and materials” has added into the cover letter (in the line 25). 

Author’s response: Thank you for your advice. The minimal data set has been summited including S4 File. Minimal data set and S5 File. Data codebook. 

Additional Editor Comments:

Please fill in the ethics statement box of the questionnaire with the approval by the Institutional Review Board of Hue University of Medicine and Pharmacy with the registration number: H2019/003.

Author’s response: Thank you for your comment. The ethics statement “Our study was approved by the Institutional Review Board of Hue University of Medicine and Pharmacy with the registration number: H2019/003” has updated in the cover letter (please see the Cover letter, line 15) and please help us to fill the statement in the online submission form if appropriate.

I would include the kind of study within the title.

Author’s response: Thank you for your suggestion. We have changed the title to “Associated factors with Premenstrual syndrome and Premenstrual dysphoric disorder among female medical students: A cross-sectional study” in line 1.

Regarding the aim of the study in the abstract, I would rephrase:

"The study aimed at determining potential risk factors associated with Premenstrual

Syndrome (PMS) and Premenstrual Dysphoric Disorder (PMDD)."

Author’s response: Thank you for pointing that out. We updated our aim to “The study aimed to determine potential risk factors associated with Premenstrual Syndrome (PMS) and Premenstrual Dysphoric Disorder (PMDD).” in line 141.

As a cross sectional study I would rater talk about associated factors and potential risk factors. Do not try to make them look causal.

Author’s response: Thank you for your suggestion. We acknowledge the limitations of the study. In the line 602, we also stated the inability to determine a causal relationship that “Second, the cross-sectional study design does not allow us to identify the causal relationship of the factors and the outcome, which makes the results prone to reverse causation”. 

The main limitation apart from the cross sectional character of the study is the weak external validity, meaning that the studied population is probably not representative of the Vietnamese female population aged 18-45 years.

Author’s response: Thank you for your great comment. We have additionally stated this limitation of the study that “First, the sample size was small and only restricted to female medical students which is not representative of Vietnamese women aged 18-45; therefore, the generalizability of the study results to all Vietnamese women is weak” in line 600.

This should be taken into account and of course displayed as a limitation in a more strong manner. ( The authors say "limits the generalizability", when they should straightly say, "results are not generalizable to the vietnamese female population aged 18-45."

Author’s response: Thank you for your comment. We have further acknowledged the limitations of the study in a stronger manner. At the line 602, we stated that “Second, the cross-sectional study design does not allow us to identify the causal relationship of the factors and the outcome, which makes the results prone to reverse causation”. 

s

Line 58, what geographic area are the authors talking about?

Author’s response: Thank you for your question. We have added further details as follows “In a meta-analysis including 17 studies, prevalence of PMS ranged from 12% (in French) to 98% (in Iran), with a pooled estimate of 47.8% (95% CI: 32.6-62.9%) (4–6). PMDD is a severe disorder of PMS affected 3 – 8% of reproductive age women verified by daily record of severity problems (DRSP) (1).” in line 184.

Line 85 please specify that the study aim is targeted to a vietnamese female medical students sample.

Author’s response: Thank you for your comments. We have updated the aim of the study that “This study aimed to determine potential risk factors associated with PMS/PMDD among the Vietnamese female student aged 18 – 45 via C-PASS or PSST” in line 291.

Figure 1

Among 447 screened women there was not a single case of endometriosis ( according to the literature around 10% of women suffer it ) please discuss as a limitation.

Author’s response: In this small sample of young women (94% under 30), we think it is expected not to observe any self-reported cases of endometriosis. As you advised, we also make it clear that our results were drawn from our study sample only, not the entire female population of Vietnam. We have added in the limitation section that First, the sample size was small and only restricted to female medical students which is not representative of Vietnamese women aged 18-45; therefore, the generalizability of the study results to all Vietnamese women is weak” in line 600 and “Third, our exclusion criteria on having any history of mental health disorders were self-reported, which might be underdiagnosed or underreported.” in line 604.

Reviewers' comments:

Reviewer's Responses to Questions

Comments to the Author

1. Is the manuscript technically sound, and do the data support the conclusions?

Reviewer #1: Partly

Reviewer #2: Partly

2. Has the statistical analysis been performed appropriately and rigorously?

Reviewer #1: Yes

Reviewer #2: Yes

3. Have the authors made all data underlying the findings in their manuscript fully available?

Reviewer #1: Yes

Reviewer #2: Yes

4. Is the manuscript presented in an intelligible fashion and written in standard English?

Reviewer #1: Yes

Reviewer #2: Yes

5. Review Comments to the Author

Reviewer #1:

 1. Title:

Number of appropriate words and is related to the objectives of the text and the development of the text.

The main objectives of the study are described.

Author’s response: Thank you for checking the appropriateness of our title and objectives. 

2. Abstract:

It describes how the study was carried out, including the methodology, without delving into the methodological details.

Author’s response: Thank you for thoroughly checking our abstract.

It is recommended that abbreviations not be included in the abstract. Abbreviations appear twice, in the abstract and in the text, remove them from the abstract.

Author’s response: Thank you for your recommendation. The abbreviations had removed from abstract (from the line 141 to the line 161).

The most important results were summarized with their interpretation.

The abstract does not exceed 300 words.

Author’s response: Thank you for your checking the word limits of our abstract.

3. Introduction:

It is recommended to write more than the types of treatment used for each disease.

Author’s response: Thank you for your suggestion. We have added further details on types of treatment in the Introduction section as follows “As the pathogenesis is still poorly understood, treatment focuses mainly on mitigating symptoms via using medications. The first-line treatment is to use serotonergic antidepressants (selective serotonin reuptake inhibitors (SSRIs))to modulate serotonin level. Other drug options are GnRH agonists or estrogens, which are considered endocrine therapies to suppress ovulation. However, these drug treatments carry worrisome side effects;the patient needs to consult with physicians before initiating treatment (12,27) . Besides, American Association Family Physician (AAFP) emphasized that eliminating modifiable risk factors can improve the severity of PMS and PMDD (10,28).” in line 208.

4. Methods:

Place the name of figure 1 and briefly explain the main losses of participants during the writing in the text.

Author’s response: Thank you for pointing this out. We have updated the Figure 1 with the name “Figure 1: Flow chart of participant recruitment and retention.” in the File “Figure 1”. In addition, we have briefly explained that “Among 447 female students who registered to participate in our study, 428 students were eligible and completed the baseline assessment. We excluded 19 patients because of having irregular menstruation (12 students) and a history of endocrine conditions within the last six months (7 students). After the follow-up phase, 302 participants (70.56%) who completely reported symptoms of at least two consecutive menstrual cycles were included in the data analysis.” from the line 446 to the line 451.

Inclusion and elimination criteria are missing in the literature review. From what year to what year was the bibliography used, what types of text were used and in what search engines did they find them. 

Author’s response: Thank you for your comment. We conducted the literature review in Pubmed library using this Boolean search term: 

((premenstrual syndrome) OR (premenstrual dysphoric disorder) OR (premenstrual disorder) OR (menstrual disorder)) AND (((Premenstrual syndrome screening tool) OR (Daily record of severity of problems) OR (Carolina Premenstrual Assessment Scoring System)) OR ((association factors) OR (caffeine) OR (alcohol) OR (menarche) OR (depression) OR (Blood group) OR (lifestyle) OR (smoking) OR (cigarette) OR (physical activity) OR (exercise)))

The article from 2014 to 2019 had been reviewed (139 articles). The syntax has been presented in the S3 File. Literature search syntax. 

5. Results:

The most important results are highlighted in the text, the tables and figures coincide with what is cited in the text.

Author’s response: Thank you very much for checking our result.

6. Conclusion:

The conclusion does not appear in the text.

Author’s response: Thank you for your suggestion, we have added conclusion at the end of the main text as follows “PMS/ PMDD are relatively common disorders in our study. The prominent risk factors for PMS and PMDD were negative Rhesus blood type, menarche age, caffeine consumption, and self-reported depression. The Vietnamese PSST was shown as an effective screening tool for PMS/PMDD with strong validity compared to the gold standard C-PASS. Monitoring the burden of these conditions over time and determining both modifiable factors that can alleviate or exacerbate the symptoms would be important for managing health of not only women but also their social connections.” in line 615. 

7. References:

The references are correctly cited in the Vancouver format and in the correct order.

Excellent study, with great impact on women to know the risk factors associated with Premenstrual Syndrome and Premenstrual Dysphoric Disorder, the objectives proposed at the beginning of the study were completed, with adequate methodology that explained step by step how they approached the study. The results were adequately expressed and an excellent review of the literature was carried out. I recommend this article for publication with the minor revision.

Author’s response: Thank you for your generous compliment.

Reviewer #2: 

The study aims to determine possible risk factors for PMS/PMDD in a Vietnamese specific population of students based on previous research about the topic in general population. The methodology is thoroughly described, with several steps, and focused on statistical analysis of the factors included in the study and in the validation of screening and diagnostic tools. Diagnostic process is adequately proposed.

Evaluating it at a whole, the statistics and description of results are thorough, concrete and described. The diagnostic process is clear with some specifications. What might be proposed to review could be the references to the population of the study and the selection of the sample.

Specifying it into notes:

1. Introduction

- The motivation of the study is adequately presented at the introduction. However, the definition and specific risk factors already studied are not clearly addressed and a thorough description could be more accurate as is the reason for the study. PMS/PMDD are sometimes referred as interchangeable, and it is necessary to specify whether evaluation tools are used for diagnosis or screening, since the guidelines set some recommendations about it. I would suggest that references from the International Society of Premenstrual Disorders (O’Brien et al) about consensus in diagnostic criteria could be included.

Author’s response: Thank you for pointing this out. 

Regarding risk factors, we have added into the Introduction section that “Because of a relatively high validity (8–10), PSST has been recommended as PMS / PMDD screening tool by International Society for Premenstrual Disorders (ISPMD) (10–13).” in line 190.

 We also added that “Among many other validated techniques, the most commonly used and accepted tool for diagnosing PMS/ PMDD is DRSP (16). In order to efficiently summarize results from DRSP, Eisenlohr-Moul et al developed an algorithm called the Carolina Premenstrual Assessment Scoring System (C-PASS) (17) that analyzes DRSP data in a standardized manner” in line 194. We have discussed these factors in details in the Discussion section in line 552 to 590.

We have read the article (O'Brien et al.) with interest and strongly agreed with the reviewer's suggestion. We have cited ref #13: “O’Brien PMS, Bäckström T, Brown C, Dennerstein L, Endicott J, Epperson CN, et al. Towards a consensus on diagnostic criteria, measurement and trial design of the premenstrual disorders: the ISPMD Montreal consensus. Arch Womens Ment Health. 2011 Feb;14(1):13–21” as the reviewer’s suggestion (please see in line 190).

2. Methods

- The population indicated (women between 18 and 45 years old) might not represent appropriately the population the study is focused on, and so the conclusions might not match with the aim of the study. University students from a specific region in Vietnam could represent the sample and population. Furthermore, prevalence might change, as it has been described higher prevalence among students and women in their twenties.

Author’s response: Thank you for your comments. We have acknowledged the limitation in a stronger manner. Most of the female student in our study were under 24, and the age distribution in our study sample was different from the Vietnamese population. Therefore, our results are only limited to the study sample. We have additionally stated this limitation of the study that “First, the sample size was small and only restricted to female medical students which is not representative of Vietnamese women aged 18-45; therefore, the generalizability of the study results to all Vietnamese women is weak” in line 600.

- Exclusion criteria are described and, as indicated in the algorithm of the study, only few of them make a true exclusion. Some studies refer more specifically to women taking antidepressants, define more concretely diseases or conditions needed to be considered in this setting and evaluate mental disease before consider inclusion as it might be underdiagnosed or difficult to determine once the diagnosis of PMS/PMDD is considered. 

Author’s response: Thank you for your insightful comments. All the health conditions were self-reported by the participants. We have acknowledged this limitation in our limitation section as follows “Third, our exclusion criteria on having any history of mental health disorders were self-reported, which might be underdiagnosed or underreported” in line 604.

- Sample size is calculated based on reported prevalence of PMS (not the described in Vietnamese population -10%-), but not PMDD (much lower prevalence described). Furthermore, the prevalence is established based on diagnostic criteria, with variable match in case of screening tools (PSST), which is the test used for the sample in the study.

Author’s response: Thank you for your comments. The PMS/PMDD prevalence of 10% in the reference #27 is our very own preliminary report based on the same dataset we report in this manuscript. 

Regarding sample size calculation using PMS prevalence but not PMDD prevalence, when we first planned to conduct this study, we used PMS prevalence of 30% reported in reproductive women from our literature review (Reference #28). PMDD is a severe version of PMS by definition. In this case, we never considered PMDD as a separate outcome but combined it with PMS. Therefore, we believe it is not necessary to calculate sample size for PMDD alone.

Regarding sample size calculation based on PSST, we used this because its previously reported sensitivity and specificity were lower than C-PASS; therefore, it should require higher sample size than C-PASS.

Reference #28: Baker LJ, O’Brien PMS. Premenstrual syndrome (PMS): a peri-menopausal perspective. Maturitas. 2012 Jun;72(2):121–5.

- Cases considered from DRSP are referred to a psychiatrist but there is no reference about whether they were confirmed or excluded. 

Author’s response: Thank you for your comment. We confirmed that all four participants screened positive for PMDD using C-PASS were referred to psychiatrists at the Psychological Department of HueUMP. Psychiatrists also confirmed all of these four PMDD cases. We have added to the Result section that “Four patients diagnosed with PMDD based on C-PASS was referred to psychiatrist and confirmed to have PMDD.” in line 469.

3. Results

- As I observe at Table 1, more than half of the sample (or approximately half) reported consuming caffeine once or less per month. 

Author’s response: We confirmed that this result is correct.

Also, we have renamed the categories to make it clearer as follows:

● Once a month or less

● 2-3 times per month to 1-3 times per week

● from 4 times per week and above

We have changed the text accordingly in all the table, figures, and text

4. Discussion

- An evaluation of why there is a change of sensitivity of PSST from pre to re-test would be interesting.

Author’s response: Thank you for your suggestion. We also found this interesting. Because these emotional and physical symptoms, social relationships are sensitive to time, we think the change in sensitivity of PSST is expected. As this paper aimed to determine associated factors with PMS/PMDD, we want to focus on discussion related to potential associated factors instead.

- The conclusion that attaches specifically to the aim of the study, is not at the end of the article.

Author’s response: Thank you for pointing this out. We have added a conclusion at the end of the main text as follows “PMS/ PMDD are relatively common disorders in our study. The prominent risk factors for PMS and PMDD were negative Rhesus blood type, menarche age, caffeine consumption, and self-reported depression. The Vietnamese PSST was shown as an effective screening tool for PMS/PMDD with strong validity compared to the gold standard C-PASS. Monitoring the burden of these conditions over time and determining both modifiable factors that can alleviate or exacerbate the symptoms would be important for managing health of not only women but also their social connections.” in line 615.

6. PLOS authors have the option to publish the peer review history of their article (what does this mean?). If published, this will include your full peer review and any attached files.

Do you want your identity to be public for this peer review? For information about this choice, including consent withdrawal, please see our Privacy Policy.

Reviewer #1: No

Reviewer #2: No

Best regards,

Ngo Dinh Trieu Vy, MD

---

## [Editor Report · Decision Letter 1]

21 Nov 2022

Associated factors with Premenstrual syndrome and Premenstrual dysphoric disorder among female medical students: A cross-sectional study

PONE-D-22-24425R1

Dear Dr. Ngo Dinh Trieu,

We’re pleased to inform you that your manuscript has been judged scientifically suitable for publication and will be formally accepted for publication once it meets all outstanding technical requirements.

Kind regards,

Ignacio Garitano Gutierrez, MD, MSc

Academic Editor

PLOS ONE
---

## [Editor Report · Acceptance letter]

1 Dec 2022

PONE-D-22-24425R1 

Associated factors with Premenstrual syndrome and Premenstrual dysphoric disorder among female medical students: A cross-sectional study. 

Dear Dr. Ngo Dinh Trieu:

I'm pleased to inform you that your manuscript has been deemed suitable for publication in PLOS ONE. Congratulations! Your manuscript is now with our production department. 

Kind regards, 

on behalf of

Dr. Ignacio Garitano Gutierrez 

Academic Editor

PLOS ONE